# Using Optical Coherence Tomography in Plant Biology Research: Review and Prospects

**DOI:** 10.3390/s25082467

**Published:** 2025-04-14

**Authors:** Ghada Salem Sasi, Adrien Alexis Paul Chauvet

**Affiliations:** School of Mathematic and Physical Sciences, University of Sheffield, Sheffield S3 7HF, UK; gssasi1@sheffield.ac.uk

**Keywords:** optical coherence tomography, plant, non-invasive, in vivo, real-time

## Abstract

Visualizing the microscopic structure of plants in vivo, non-invasively, and in real-time is the Holy Grail of botany. Optical coherence tomography (OCT) has all the characteristics necessary to achieve this feat. Indeed, OCT provides volumetric images of the internal structure of plants without the need for histological preparation. With its micrometric resolution, OCT is commonly used in medicine, primarily in ophthalmology. But it is seldom used in the field of botany. The aim of the present work is thus to review the latest technical development in the field of OCT and to highlight its current use in botany, in order to promote the technique and further advance research in the field of botany.

## 1. Coming of Age

Optical coherence tomography (OCT) is, at its core, an interferometric imaging technique. One of the first uses of low-coherence interferometry induced by backscattering within soft tissues was by Fercher et al. in the 1980s [1]. In this pioneering work, Fercher et al. succeeded in measuring the optical length of the human eye in vivo. After nearly a decade of technical development, the first OCT setup was demonstrated in 1991 [2,3,4,5] by Fujimoto et al. In comparison to Fercher’s single-point imaging, OCT provides cross-sectional and three-dimensional imaging of soft tissues while remaining noninvasive [6]. Five years later, the first commercial OCT instrument was launched [7] and was rapidly implemented for healthcare applications. Today, in light of its imaging capabilities and practicality, OCT is routinely used by ophthalmologists to examine patients [8]. However, using OCT imaging as a non-invasive and real-time technique to investigate plants tissue remains rare. Its first use in the field of botany was reported by Sapozhnikova et al. in 2004 [9]. In this work, the team described their ability to directly visualize the dehydration and rehydration dynamics of *Tradescantia pallida* (Rose) leaves. The penetration depth was about 1–2 mm and the spatial resolution was about 15 μm [9]. OCT techniques have, since then, greatly improved in terms of scan rates, contrast, sensitivity, and phase-stability [2,10]. For example, from an initial~400 axial scans (A-scans) per second [11], OCTs are now commonly running at hundreds of kHz [12], thus improving spatial and time resolution. These improvements in acquisition speed and precision are paving the way for real time functional imaging in all scientific fields.

Time-domain OCT (TD-OCT), which was the first type of system, was using a moving reference mirror to capture images, but it was slow and had limited resolution. Fourier-domain OCT (FD-OCT), which was later developed, instead uses Fourier transformation, thus improving speed and image quality. The development of swept-source OCT (SS-OCT), in particular, further enhanced imaging depth and speed by making use of novel tunable lasers. All these advancements make OCT ready for agricultural applications, such as effective plant health monitoring and food quality assessment [11,13,14,15].

## 2. Concepts and Characteristics of OCT

The common feature of OCT systems is their ability to generate non-invasive, high-resolution, depth-resolved images of soft tissue solely by collecting backscattered light [16,17]. This backscattered light is then made to interfere with a reference light originating from the same initial light source to produce an interferogram, be it a fixed wavelength laser, a swept-source laser, or a superluminescent diode [18]. And since backscattering is the result of a variation in refraction index between differing intrinsic structural components, analysis of the interferogram allows the retrieval of this depth-resolved structural information.

In comparison to the narrow light sources and long coherence lengths typically used in Michelson interferometry [19], OCT benefits from spectrally broadband light sources with low temporal coherence and high spatial coherence, which improves the depth resolution of the resulting images [14]. The sample arm of this low coherence interferometer is often extended via an optic fiber for hand-held applications. This feature enables the OCT systems to be adapted to target samples, which is advantageous for in vivo biological applications.

OCT can be performed either in the time or spectral domain. On the one hand, as introduced earlier, TD-OCT produces tomographic images by oscillating a reference arm mirror of the interferometer to create temporal interferences. On the other hand, Fourier-domain OCT (FD-OCT) generates the spectral interferogram by mixing the broadband backscattered light from the sample with the reference light without changes in the reference arm length [20]. Because the core of FD-OCT is devoid of moving parts and because Fourier analysis of spectral interferogram is fast, the technique has the advantage of being ~100 times faster than TD-OCT [21,22] while being more robust for in-field applications. The FD-OCT family includes both the Sprrctral domain OCT (SD-OCT) as well as SS-OCT. While SD-OCT still uses a spectrometer with a dispersive element and a relatively slower detector array, the SS-OCT benefits from an optical source which rapidly sweeps a narrow linewidth over a broad range of wavelengths. In SS-OCT, each wavelength is then detected sequentially during each sweep by a high-speed photodetector. This technical feat allows SS-OCT to reach hundreds of kHz in image acquisition rates, thus increasing in reliability when monitoring moving objects [23]. Note that when it comes to scanning speed, typical TD- and SD-OCT still requires scanning of either the reference mirror or of the excitation wavelength, respectively, to produce an entire b-scan. This scanning requirement can be bypassed in “single-shot” OCT systems, which can acquire full b-scan at every shot by using a broad light source and a diffracting element to record each wavelength separately and simultaneously via photodetector arrays [24]. However, even if each b-scan is taken virtually instantaneously, the processing of each scan might take up to 140 ms [25].

In term of image quality, the axial resolution (Z-axis) typically depends on the bandwidth of the OCT light source; the lateral resolution (X-Y plane) is usually given by the numerical aperture of the microscope objective used, as depicted in Figure 1. Resolutions of both axis are typically in the order of 1–10 µm in OCT systems [26,27]. The sensitivity, on the other hand, is related to a combination of factors, including laser power (typically in the µW range to avoid damaging the tissues), scattering [28,29] (which can be enhanced with contrast agents), and scanning speed (of up to MHz for FD-OCTs).

Overall, OCTs are technically simple, compact, robust, and adaptable to various environments, which make them ideally suitable for biological applications.

## 3. Popularity of OCT in Botany

Owning to its many advantages, OCT has been extensively utilized in the medical field for various health applications and, more specifically, in ophthalmology, cardiology, and dermatology. The widespread adoption of OCT is illustrated by the ever-increasing number of publications pertaining its use in the medical field, numbering in the tens of thousands every year, as shown in Figure 2. In comparison, following the same metric, the use of OCT in botany is about fifty times less prevalent. This wide disparity between the use of OCT in the medical field and in botany is actually surprising given the comparable suitability and potential impact this technique has on both fields.

## 4. Comparison with Alternative Techniques

To better appreciate the potentialities of OCT in plant biology research, it is useful to compare it with common imaging techniques that are readily used in the field. And beyond the various technical characteristics of each imaging tool, the following discussion also includes practicality and ease of access, which is where OCT’s primary strengths reside.
The highest resolution technique available is X-ray tomography (XCT). This system provides 3D rendering of the internal structures of plant tissue with nanometric resolution [30,31]. Samples can be as large as a whole plant (e.g., 40 cm tall), depending on the size of the sample chamber, but it is not field-applicable [32]. Furthermore, high X-ray dose causes ionization, which disrupts and damages the plant [33,34,35].Hyperspectral imaging (HIS), which includes UV reflectance imaging [36,37], serves as a non-invasive and efficient tool for studying plants and whole crops. Although such spectroscopic method is in theory diffraction limited, HIS’ resolution is typically low given the sensor size and its distance from the object [38]. Although HIS typically has the lowest spatial resolution [39], it is the most practical for field application, offering insights into the health, physiology, and interactions of plants with their environment.Raman microscopy is often used as a convenient and non-invasive way to monitor the presence of specific molecules in tissues. Molecules are categorized by the vibrational signature of their functional groups [40]. For this technique, no sample preparation is needed; it is non-destructive and is highly sensitive, which makes it field-applicable. The past decade has seen the development of universal multiple angle Raman spectroscopy (UMRAS) for monitoring functional groups of embedded molecules, thus paving the way for 3D Raman imaging [41,42]. But to characterize tissues solely based on the functional groups of its constituting molecules is not at all trivial, and variations among research teams makes it difficult to obtain replicate measurements [43].Laser-induced fluorescence (LIF) is commonly used, like HIS, to remotely assess whole leaves, plants, and even crops [44]. It allows for real-time imaging and is non-invasive [45]. LIF typically gives information about the presence of chlorophyll via its induced fluorescence [45]. Consequently, LIF is restricted to monitoring chlorophyll and other highly fluorescent molecules within tissues, without revealing direct information about the tissue’s internal structure.Magnetic resonance imaging (MRI) is another non-invasive, non-destructive imaging technique, which also provides three-dimensional images. It also allows for whole plant investigation [46] and is non-invasive [47]. However, MRI has a typical axial resolution of 1.5–2.0 mm [18,48] and is not field-applicable.Ultrasound imaging is another 3D imaging technique which uses sound instead of radiation. It can be used to investigate plant tissue and water movement within it [49]. Ultrasound imaging benefits from a significantly higher penetration depth compared to OCT, typically up to several centimeters depending on the frequency used. However, it has lower axial and lateral resolution compared to OCT [50], often around 10 to 100 times lower, ranging from about 50 µm to 500 µm [51] while OCT’s axial resolutions typically range from 1 to 15 µm [49,50,51]. Ultrasounds have been shown to directly affect plants, albeit in a positive way [52].

Hence, in light of the conventionally used imaging techniques, it is understood that OCT represents a powerful compromise between resolution and practicality, bringing 3D imaging into the field for real-time analysis.

## 5. The Various Variants of OCT

The suitability and potential impact of OCT in the field of botany can be further appreciated by exploring the different types of OCT systems that have already been employed in botany. Benefiting from its technical simplicity (compared to alternative imaging systems), OCT is highly suitable for a range of multimodal imaging techniques:

### 5.1. Polarization-Sensitive OCT

Polarization-sensitive OCT (PS-OCT) benefits from the ability of fibrous structures in altering the polarization of light. Because different tissues alter the polarization of light differently, this system provides an added layer of contrast, that of polarization, on top of the typical grey-scale OCT images. For example, it allows differentiation between distinct cellular layers that would otherwise appear as the same grey-shade [53]. More specifically, it measures birefringence, which reflects tissue organization by assessing how light slows differently along specific axes, with the fast axis orientation indicating the preferred direction of light propagation within the tissue, and the degree of polarization uniformity (DOPU) helping identify regions with uniform or disrupted polarization properties.

PS-OCT can be further extended by applying the Mueller matrix method to capture diattenuation, which describes how tissue differentially absorbs or transmits polarized light. This provides additional contrast for distinguishing subtle tissue characteristics [54].

### 5.2. Full-Field Optical Coherence Tomography (FF-OCT)

Full-field OCT is a technique that differs from time-domain and frequency-domain OCT by producing tomographic images without scanning a light beam using galvanometer scanners [55,56], similar to the “single-shot” OCT system discussed earlier [24,25]. Instead, the entire sample is illuminated with a light of extremely short coherence. The tomographic images are thus obtained in the en-face orientation (orthogonal to the optical axis) by a Linnik-type interferometer. Accordingly, full-field OCT is also called en-face OCT or, more commonly, full-field optical coherence microscopy (OCM). The transverse resolution of full-field OCT is similar to that of conventional microscopy (~1 μm) but has the advantage of being non-invasive and not requiring any histological treatments. The axial resolution, determined by the spectral properties of the illumination source, is also of the order of 1 μm [57].

### 5.3. Spectroscopic Optical Coherence Tomography (S-OCT)

S-OCT differs from standard OCT by further analyzing the spectrum of the backscattered light [58,59]. Analogous to the PS-OCT, S-OCT provides additional contrasts which consist of the spectral blue- or red-shift of the backscattered light’s maximum amplitude. Accordingly, by monitoring the spectrum of the scattered light, structures that selectively absorb part of the incident light can thus be distinguished.

### 5.4. Biospeckle OCT (bOCT)

Biospeckle imaging consists of analyzing the backscattered light from a coherently illuminated object. Typically, for static objects, the backscattered rays interfere with themselves and create a speckle pattern that is unique to the object’s surface and internal structure (depending on the penetration depth of the light used). If the objects are biological samples, the flow of their constituent parts (e.g., flow of red blood cells through veins [60]) will generate a continuously evolving speckle pattern called biospeckle. Biospeckle OCT (bOCT) is also known as “dynamic OCT” [61], as well as “OCT angiography” or “optical coherence angiography” [62]. Monitoring biospeckles over time helps assess the level of physiological activity of the sample. Thus, in comparison to OCT, bOCT provides an additional contrast which consists of the physiological activity of the sample.

### 5.5. Inverse Spectroscopic OCT (ISOCT)

In this variant of OCT, the signals are further analyzed to extract depth-dependent absorption and scattering parameters [63]. Indeed, attenuation in OCT describes the weakening of the light signal as it passes through tissue, caused by both absorption and scattering. The more attenuation there is, the weaker the OCT signal becomes, which can affect the clarity and quality of the resulting images [64]. Thus, by monitoring the changes in scattering intensities, it then becomes possible to infer about the concentration and size of the scattering particles at various depths.

While many more types of OCT are currently employed in healthcare, the few that have been selected here are among those that have been demonstrated in botany in only the last decades, showing that the field is fast developing.

## 6. Use of OCT in Botany

As already alluded to, the use of OCT in ophthalmology has grown to the point where it is now an indispensable tool in the field. But OCT’s abilities to investigate in real-time soft tissues without histological preparations, and without incurring radiation damages, are equally suited for botany. Plants, like skin and retinae, are comprised of soft tissues that can be subject to the exact same type of investigations, as demonstrated subsequently.

### 6.1. OCT for Non-Invasive Investigation of Crop Types

Because OCT uses scattered light to probe the internal structure of tissues, the depth and overall quality of the images depend on the type of tissues investigated. For example, while OCT can image the whole cross section of a soft Arabidopsis leaf [65,66,67], the same equipment is barely able to image the first cell layer of a sturdy tomato leaf due to insufficient light penetration [68]. Indeed, the denser the sample, the greater the absorption and the less scattered light is collected from the sample’s internal structures. This phenomena is illustrated in J.C. Clements’ work on thick lupin seeds [69]. In this study, although OCT was used to distinguish between different species by looking at differences in hull thicknesses, as shown in Figure 3, the penetration depth was limited to ~200 μm with no discernable structural element within specific layers.

On the other hand, water-rich samples, such as onions [70], kiwi, and orange fruits, [67] allow the visualization of individual vacuoles up to 1 mm under the surface, as depicted in Figure 4. It is, however, important to note that penetration depth within tissues is wavelength-dependent. But overall, near-IR is ideally suited for plant investigations [71].

In another example, OCM has been used to investigate the soft Arabidopsis plant [72,73]. Along with the vacuoles, OCT was successful in visualizing other subcellular structures such as the trichomes’ nuclei and organelles. OCT was similarly successful in distinguishing distinct stages in the senescence of leaves by monitoring texture changes at differing yellowing stages [74].

Although dense samples typically yield limited imaging depth, complementary image processing can help circumvent the limited resolution and extract additional information such as particle size and concentrations. For example, ISOCT has been used to estimate depth-resolved concentrations of chlorophylls in corals [75].

Overall, by providing details of internal structures, OCT has been demonstrated to be a fast and reliable tool to investigate and differentiate between crop types.

### 6.2. OCT to Study Plant’s Responses to Biotic Stresses

Although OCT’s typical micrometric resolution does not allow for direct visualization of fungi, it can still be used to differentiate between infected and non-infected crops whenever the infection alters the plant’s overall internal structure. For example, OCT has been used to study the morphological changes in response to *Anthracnose* [76], a fungal disease that typically causes dark lesions on leaves [77]. It has been used to follow the development of a progressive rind breakdown disorder in ‘*Nules clementine*’ mandarin [65,66,67] via the progressive collapse of oil glands. Similarly, OCT has been used to detect pathological infections such as *Venturianashicola*, which causes pear scab disease [78,79,80]. It has also been used to investigate the gray leaf spot disease in *Capsicum annuum* leaves [81], to diagnose *marssonina* blotch disease in apple leave [82], to detect melon seeds infected with the Cucumber green mottle mosaic virus [83,84], to investigate fungal (*Botrytis allii*) and bacterial (*Pseudomonas* sp.) infection in onions [82,85], to detect Anthracnose fungus-infected tomato seeds [86], to investigate virus-infected orchid plant leaves [87,88], and to detect defects and rot in onions [80,85]. Considering these various applications, OCT has demonstrated to be an ideal diagnostic tool for the quality control of crops, allowing for early treatment and reducing waste.

### 6.3. OCT to Study Plant’s Responses to Abiotic Stresses

Beyond the visualization of plants’ morphological changes induced by infections, OCT can effectively monitor the effects of environmental changes. For example, OCT was used to study the effect of drought and rehydration on leaf morphology [89], as well as to study the effect of ozone stress on Chinese chives (*Allium tuberosum*) leaves [90]. Similarly, the advantages of OCT over more advanced techniques, such as confocal microscopy and micro-CT, were further demonstrated while monitoring the effects of preharvest fertilization treatments and of fruit storage on fruits such as apples [91] and kiwi [92].

These last examples illustrate that OCT is also suited to studying the effects of environmental factors incurred through urbanization and industrialization.

### 6.4. OCT for Investigation of Live Responses

Since OCT measurements are non-invasive, successive measurements can be performed at the same location without altering the plant’s metabolism and physiological functions. This characteristic enables the investigation of time-resolved dynamics of structural changes within a specific section of the plant. And given the fast acquisition speed of OCT, the time laps between measurements can be easily adjusted to the process under investigation.

In this manner, the internal changes induced by the rotting of apples could be followed over the course of 25 days [80]. The spread of rot was evidenced by a gradual increase of the infected region in a lateral direction and a gradual disappearance of the cuticle, wax, epidermis, and hypodermal layers. In another instance, OCT was successfully used to monitor the emergence of roots from switchgrass seeds over a lap of 21 h [93,94]. Similarly, the effect of ozone on the internal cell structure of Chinese chives leaves was investigated via bOCT [95,96]. The technique was effective in distinguishing changes only a few hours after ozone exposure.

In faster timescales, OCT was also used to monitor modifications in the distribution and structure of chloroplasts in tobacco, only minutes after inoculation of the bacterial elicitor harpin protein [97].

Finally, through continuous monitoring, OCT was used to investigate signaling mechanisms in real time, as shown in Figure 5. It was then possible to follow the propagation of slow wave potential, induced by laser burn, across a young tomato plant [98].

## 7. The Future of OCT

Although the vast majority of research involving OCT still pertains to the medical field, this brief review demonstrates the suitability of OCT in botany as well. In comparison to current imaging tools, OCT suffers from a resolution which seems limited to ~1 µm, regardless of the equipment type. The technique, however, hugely benefits from its simplicity, real-time acquisition, and non-invasiveness. OCT is thus prone to develop in research fields that require site-specific, systematic and/or real-time monitoring. As such, OCT is ideal for systematic screening of fruits and vegetables for quality control purposes. It is also ideally suited for real-time monitoring of responses to biotic and abiotic stresses. It is noted that across the board, the quality of OCT images is often speckled, which is a common issue in coherence-based imaging techniques [99]. Fortunately, with the advent of machine learning algorithms, while speckle noise can be treated directly [100], further qualitative information can be more efficiently and effectively extracted by automated image processing such as registration and segmentation [101].

Given the simplicity and robustness of the equipment, it is a matter of time until OCT is applied for field experiments and finds its place in a farmer’s toolbox for direct crop screening. Hand-held OCT are currently being developed [93,94,102]. The impact of a portable OCT is huge and is attracting attention, as illustrated by the funding of projects aimed at its development [103], even if the vast majority of these projects are still dedicated to clinical purposes.

## Figures and Tables

**Figure 1 sensors-25-02467-f001:**
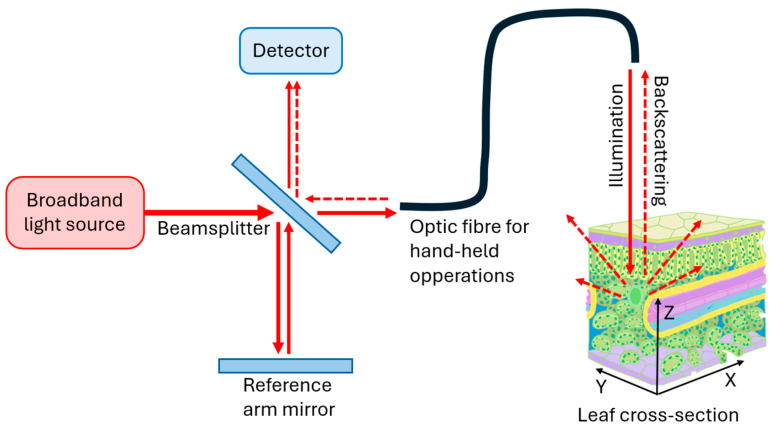
Principle of interferometry with beam propagation through optic fiber for hand-held applications. The sample is scanned laterally (X–Y) through sets of galvanometers. Created by the authors.

**Figure 2 sensors-25-02467-f002:**
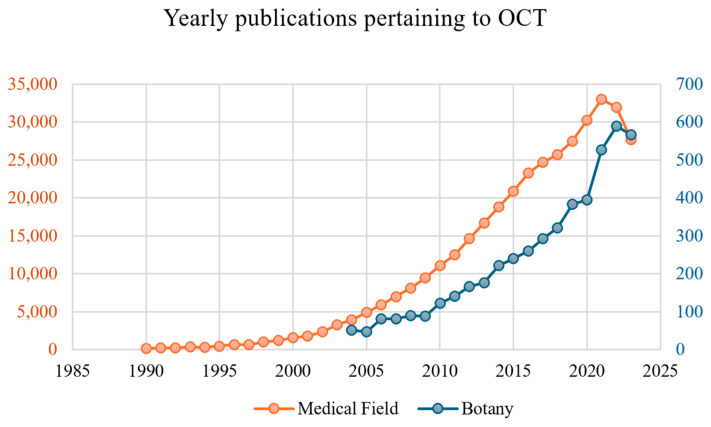
Annual number of OCT-related publications (blue) compared to the number of publications specific to the field of OCT applied to botany (In medical field: using keyword search: “optical coherence tomography” OR medical field. In botany: using keyword search: “optical coherence tomography” botany OR plant OR leaf OR seed OR plant’s root).

**Figure 3 sensors-25-02467-f003:**
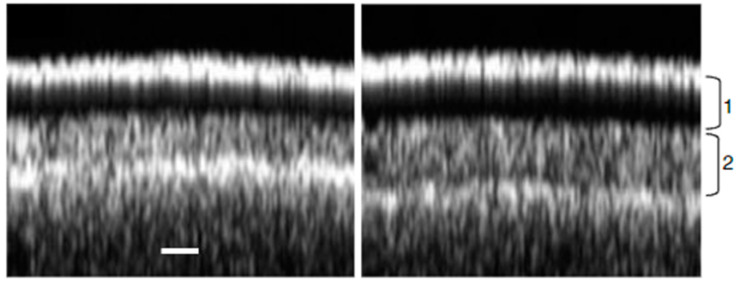
OCT images show the layers of lupin seeds. (1) is the first palisade layer. (2) is the second palisade layer. The scale bar is 50 μm. Figures were adapted from [69].

**Figure 4 sensors-25-02467-f004:**
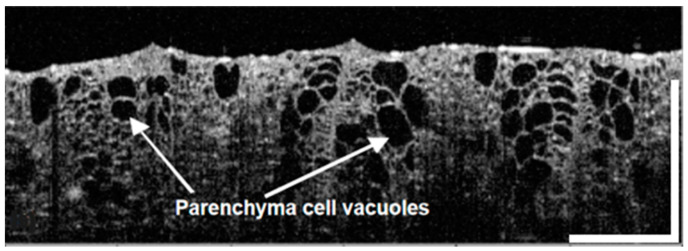
An OCT image of the kiwifruit parenchyma cell vacuoles. Figure adapted from [67]. Scale bars are 1 mm.

**Figure 5 sensors-25-02467-f005:**
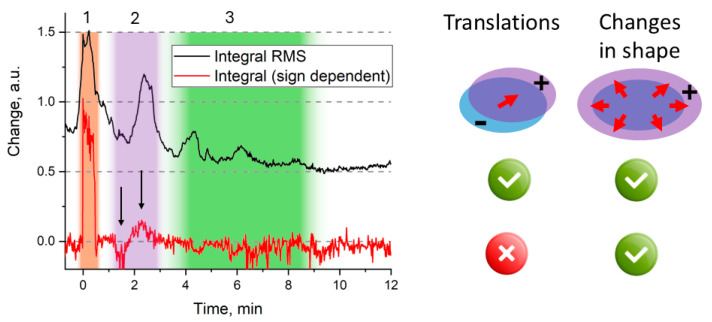
Time-resolved integrated A-scans difference image of a tomato leaf after laser burn. The red curve represents the integral accounting for changes in sign (normalized), while the black curve indicates the magnitude of the squared root of the integral (normalized and adjusted vertically for clarity). Each integral corresponds to specific cellular changes depicted to the right. The feature observed between 0 and 0.5 min is partially attributed to light scattering caused by a 30 s wounding laser pulse (zone 1). The shaded purple area (zone 2, black arrows) denotes the anticipated changes in the shape of the adjacent leaf. The shaded green area (zone 3) denotes the translational relaxation of the leaf. Adapted from [98].

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
