# Peer review of "Using Optical Coherence Tomography in Plant Biology Research: Review and Prospects"

_sensors, 2025, doi:10.3390/s25082467_

Round 1
Reviewer 1 Report
Comments and Suggestions for Authors
I praise the authors for the timely review. I want the authors to address the issues listed below:
1.Authors describe about different type of OCT; please include a description about the principles of Swept source OCT which is practical and highly prevalent nowadays.
2. Many places references are not rightfully sighted because of the reference linking. Please fix them; for this point I cannot really check whether the references sited are appropriate.
3. Authors talk about different modalities in imaging. It would be nice to have some discussion about the different modalities, the signals obtained and the interpretation of signals. Such a discussion would be helpful for the future choice of the appropriate modality for using by the user.
4. Finally, this could be tough as the research done with OCT is only on humans and not plants. Consideringthe pros and cons of OCT, at what situations OCT is expected to become a unique tool. If you can think of anything please explain.
Author Response
Comment 1: I praise the authors for the timely review. I want the authors to address the issues listed below. Authors describe about different type of OCT; please include a description about the principles of Swept source OCT which is practical and highly prevalent nowadays.
Response 1: A section has been added, explaining the concepts and advantages of SS-OCT as a type of FD-OCT.
Comment 2: Many places references are not rightfully sighted because of the reference linking. Please fix them; for this point I cannot really check whether the references sited are appropriate.
Response 2: The references have been checked for crosslinking and duplicates.
Comment 3: Authors talk about different modalities in imaging. It would be nice to have some discussion about the different modalities, the signals obtained and the interpretation of signals. Such a discussion would be helpful for the future choice of the appropriate modality for using by the user.
Response 3: We added the suggested information where it was missing (e.g. the type of signals monitored and/or the interpretation of these signals).
Comment 4: Finally, this could be tough as the research done with OCT is only on humans and not plants. Consideringthe pros and cons of OCT, at what situations OCT is expected to become a unique tool. If you can think of anything please explain.
Response 4: OCT can become a unique tool for in field analysis. That discussion has been expanded. One example is the fast quality screening of vegetables.
Reviewer 2 Report
Comments and Suggestions for Authors
This review of various OCT techniques and their medical and botanical applications is of good research value and is recommended for publication.I just have one small question.The authors pointed out that the penetration depth of OCT in biological samples depends on the type of tissue studied, and it is undeniable that different types of tissue do have different light penetration capabilities. However, common OCT systems have a variety of wavelengths such as 850 nm, 1310 nm, 1550 nm, etc.The optical penetrability of systems with different wavelengths is also different, and it is suggested that the authors add this information and elaborate on that.
Author Response
Comment 1:
This review of various OCT techniques and their medical and botanical applications is of good research value and is recommended for publication.I just have one small question.The authors pointed out that the penetration depth of OCT in biological samples depends on the type of tissue studied, and it is undeniable that different types of tissue do have different light penetration capabilities. However, common OCT systems have a variety of wavelengths such as 850 nm, 1310 nm, 1550 nm, etc.The optical penetrability of systems with different wavelengths is also different, and it is suggested that the authors add this information and elaborate on that.
Response 1: A note about the penetration depth of light and its wavelength dependence has been added.
Reviewer 3 Report
Comments and Suggestions for Authors
In general, it is a well-structured and presented review of the topic. I have only minor comments.
1. General schematic shown in Figure 1, Is not clear. If it is spectral, a grating is missing before the detector, or change it all to appear as a spectrometer. The symbol used for the optical fiber looks weird; a single line will do the job.
2. Figure 2 appears as a very low-resolution image in the PDF.
3. Lines 75 and 76 show an error code for a cited reference.
4. In section 5, the single-shot versions of OCT (a single B scan without scanning) are missing when describing the OCT variations. They have only described the single point versions (temporal or Fourier domain) to move to the full field version (en-face). Some examples of these missing setups are: DOI 10.1088/1464-4258/2/1/304, https://doi.org/10.1364/OE.14.009643
Author Response
Comment 1: In general, it is a well-structured and presented review of the topic. I have only minor comments. General schematic shown in Figure 1, Is not clear. If it is spectral, a grating is missing before the detector, or change it all to appear as a spectrometer. The symbol used for the optical fiber looks weird; a single line will do the job.
Response 1: Fig 1 has been modified accordingly.
Comment 2: Figure 2 appears as a very low-resolution image in the PDF.
Response 2: Fig 2’s quality has been enhanced.
Comment 3: Lines 75 and 76 show an error code for a cited reference.
Response 3: References have been checked.
Comment 4: In section 5, the single-shot versions of OCT (a single B scan without scanning) are missing when describing the OCT variations. They have only described the single point versions (temporal or Fourier domain) to move to the full field version (en-face). Some examples of these missing setups are: DOI 10.1088/1464-4258/2/1/304, https://doi.org/10.1364/OE.14.009643
Response 4: A discussion related to single-shot OCT has been added, both when describing the different types of OCT and the en-face modality.